microbiology, environmental science, ecology

methane, Antarctica, microbial biogeography, microbial succession, ecosystem function

**Author for correspondence:**
Andrew R. Thurber
e-mail: athurber@coas.oregonstate.edu

†Present addresses: National Institute of Water & Atmospheric Research (NIWA), Wellington, New Zealand; and School of the Environment, University of Auckland, Auckland, New Zealand.
‡Present address: Mycotic Diseases Branch, Centers for Disease Control and Prevention, Atlanta, GA, USA.

# Riddles in the cold: Antarctic endemism and microbial succession impact methane cycling in the Southern Ocean

Andrew R. Thurber[1,2], Sarah Seabrook[1,†] and Rory M. Welsh[2,‡]

[1]College of Earth, Ocean, and Atmospheric Sciences, and [2]Department of Microbiology, College of Science, Oregon State University, Corvallis, OR, USA

ART, 0000-0003-0383-832X; SS, 0000-0002-3221-3575

Antarctica is estimated to contain as much as a quarter of earth's marine methane, however we have not discovered an active Antarctic methane seep limiting our understanding of the methane cycle. In 2011, an expansive (70 m × 1 m) microbial mat formed at 10 m water depth in the Ross Sea, Antarctica which we identify here to be a high latitude hydrogen sulfide and methane seep. Through 16S rRNA gene analysis on samples collected 1 year and 5 years after the methane seep formed, we identify the taxa involved in the Antarctic methane cycle and quantify the response rate of the microbial community to a novel input of methane. One year after the seep formed, ANaerobic MEthane oxidizing archaea (ANME), the dominant sink of methane globally, were absent. Five years later, ANME were found to make up to 4% of the microbial community, however the dominant member of this group observed (ANME-1) were unexpected considering the cold temperature ($-1.8°C$) and high sulfate concentrations (greater than 24 mM) present at this site. Additionally, the microbial community had not yet formed a sufficient filter to mitigate the release of methane from the sediment; methane flux from the sediment was still significant at 3.1 mmol $CH_4$ $m^{-2}$ $d^{-1}$. We hypothesize that this 5 year time point represents an early successional stage of the microbiota in response to methane input. This study provides the first report of the evolution of a seep system from a non-seep environment, and reveals that the rate of microbial succession may have an unrealized impact on greenhouse gas emission from marine methane reservoirs.

## 1. Introduction

The concentration of methane, the second most important greenhouse gas after $CO_2$, has increased in the atmosphere 150% since 1750, up to 1.8 ppm, without a known cause [1–3]. Among significant knowledge gaps in the global methane cycle are the reservoir volume and biological sinks of Southern Ocean and Antarctic methane. Antarctica is estimated to contain between 80 and 400 Gt C methane which is a significant proportion of, and yet not included in, the approximately 1800 Gt C methane estimated to be contained in sediment-hosted marine reservoirs [1,4,5]. No active methane seeps have been discovered in Antarctica, hindering our understanding of the processes that regulate the release of Antarctica's methane. In 2011, a methane seep formed at 78° South, providing an opportunity to identify the microbial taxa involved in Antarctica's methane cycle and simultaneously track the microbial succession following the onset of methane emission.

Atmospheric forcing by marine methane sources remains minor compared to terrestrial sources largely owing to the activity of bacteria and archaea that consume methane (methanotrophy) prior to its release from the hydrosphere [6]; however, these taxa can be very slow growing and may be unable to rapidly

**Figure 1.** Early succession of the Cinder Cones methane seep. The seep is a linear feature that extends across the 10 m isobath and marked by white, sulfide-oxidizing bacterial mats on the surface. The feature was first sampled in 2012 (left panel) and continued until 2016 (centre). The mat was slightly reduced in its surface manifestation by 2014 (lower right). In 2016, more areas of active seepage were discovered including the 7 m water depth 'Shallow Site' shown in the upper right. (Online version in colour.)

respond to changing methane emissions. Climate change will increase the release of methane from subsurface marine reservoirs and while predicting the impact of this release is multifaceted [1], methanotrophy in the oceans is expected to minimize the atmospheric footprint of this release. At established methane seeps, a consortia of ANaerobic MEthane oxidizing archaea (ANMEs) and sulfate reducing δ-proteobacteria consume an estimated 70–90% of methane released from the subsurface through the anaerobic oxidation of methane (AOM; [1,7–10]). ANME microbial consortia are slow-growing, with previous measured doubling times of two to seven months [11,12]. This slow growth rate suggests that ANME have an inability to respond to both new areas of methane release or alterations in the rate of release at existing seeps. The few studies that have quantified the population dynamics of ANME following perturbation have shown responses in as little as five weeks [13], no response for a year [14], recruitment of ANME after 48 months [15], or a response by the community after several years [16]. The majority of methane not oxidized by ANME aggregates is aerobically oxidized by a diversity of bacteria including Methylococcaceae (γ-proteobacteria) in marine systems.

The Southern Ocean and the Antarctic continent remain enigmas in regards to methane content and knowledge of which microbiota involved in the methane cycle are present and active. Only two methane seeps are known in the Southern Ocean including the recently discovered South Georgia shelf habitat [17–20] and a now extinct seep that was discovered in the Larson B region of the Antarctic Peninsula. At this later seep, ANME-3 were discovered leading Niemann et al. [21] to hypothesize members of the ANME-3 clade may be adapted to cold temperatures, a finding supported by the global distribution of ANME-3 including at sites in the Arctic [16,21–23].

Two fundamental questions exist as we aim to understand the role of Antarctic and Southern Ocean methane the earth system: (i) does the physical isolation or cold temperatures of the Southern Ocean alter the fauna or processes responsible for consuming Antarctic methane? and (ii) how fast do microbial communities respond to changes in the methane cycle? Here, we begin to address these questions following the fortuitous discovery of an Antarctic marine methane seep at a site known as Cinder Cones in McMurdo Sound within the Ross Sea. This discovery has provided an opportunity to identify the taxa involved in the Antarctic methane cycle and the succession of microbiota in response to methane emissions.

## 2. Material and methods

### (a) Site description

The Cinder Cones Seep (CCS) is on the flanks of the volcanic Ross Island (77° 47.998′ S 166° 40.241′ E). This feature is adjacent to cinder cones of Mt Erebus formed greater than 0.4 million years ago and is in an area that has elevated heat flow beyond what is expected from relic activity [24], however the water temperature is −1.8°C year round. In 2011, S. Kim (Moss Landing Marine Laboratories, personal communication) observed what appeared to be the beginnings of an expansive microbial mat at this site. In 2012, this feature had increased to a 70 m × 1 m microbial mat (figure 1). The mat was not seen in 2010 despite being a prominent feature when it appeared in 2011 and occurring at a site studied since the mid-1960s for its ecology, including as a site of an ice burg scour at deeper depths [25]. No imagery exists of the 10 m deep site until 2012. Opportunistic samples were collected via sediment cores in 2012, imagery was collected in 2014 and extensive sampling of the feature was performed in 2016 including biogeochemical characterization of the feature. A second linear microbial mat was discovered and sampled in 2016 that extended along the 7 m isobath.

### (b) Sediment core collection

To identify the microbial community, grain size and chemical environment of the CCS, sediment cores (6.4 cm diameter) were collected using SCUBA. Sampling points were randomly distributed within patches of white, putative sulfur-oxidizing filamentous bacteria in 2012 ($n = 3$) and 2016 ($n = 12$) along the

feature and purposefully including both ends of the feature to capture along feature variance. For comparison, cores were also collected within 2 m of the seep (called *reference*; n = 4) and at *control* locations that were downslope, away from seep influence at Cinder Cones and two additional sites within McMurdo sound ('Jetty': 77° 51.101′ S 166° 39.933′ E and 'Turtle Rocks': 77° 44.615′ S 166° 46.297′ E; n = 3 at all control sites which were sampled at 20 m water depth). The shallow linear microbial mat was also sampled, however with a single core so it is not included in quantitative statistical analysis. Cores were transported at *in situ* temp (−1.8°C) to McMurdo Station and sliced vertically (intervals given in figure 4) and the sediment was frozen at −80°C for microbial characterization. A subset of cores had sequential 3 cm deep subcores taken vertically for methane analysis and preserved as described below.

Pore water was collected *in situ* using Rhyzon© porewater extraction devices and analysed for methane, ion and sulfide concentrations. *Ex situ* porewater extraction and oxygen microprofiling was confounded by the high porosity of the sediment, leading to rapid pore water loss and non-reproducible results; these data are not presented. For methane, 3 ml of porewater or sediment was placed in a serum bottles with 2 ml of 5 M NaOH and capped with butyl rubber stoppers and held upside down at 4°C until analysis. In 2012, to demonstrate the presence of methane, a non-quantitative sample was collected by placing a Rhyzon in a syringe of frozen sediment and extracting the porewater as the sediment thawed. To measure hydrogen sulfide, 1 ml of porewater was preserved in 0.05 M zinc acetate and analysed spectrophotometrically [26]. Ion's were analysed on a Dionex DX-100 at California Institute of Technology. Methane samples were analysed on a HP 5890 gas chromatograph with an AllTech Porapak N8/100 column at Oregon State University. Sediment grain size was measured by freeze-drying and then shaking the sediment in nested sieves. $\delta^{13}C$ and $\delta^2H$ of methane were analysed at the University of California, Davis on a Thermo Delta V Plus isotope ratio mass spectrometer.

Methane flux was measured in 2016 through the deployment of benthic flux chambers within the seep and at the reference site. Benthic chambers were 10 cm internal diameter sediment cores capped with a lid that included a septa allowing time point sample collection without perturbation and a magnetic propeller system which enabled water mixing prior to each sample collection. Each chamber sampled 0.08 $m^2$ of the seafloor and samples were taken at 12 h intervals with a syringe through the septa. At no point were the chambers opened during the deployment. Upon recovery, the entire chamber was removed, including the sediment within it, by capping the bottom of the chamber and extracting it from the sediment including approximately the top 10 cm of sediment.

### (c) Microbial community analysis

DNA was extracted from between 0.25 and 0.5 g of sediment using the MoBio (now Qiagen) DNeasy PowerSoil kit following manufacturer protocols. Earth Microbiome Project (EMP) Protocols were followed as specified in [27], including amplification with the 515f and 806rb primers [28]. For community analysis, we used forward reads generated on an Illumina MiSeq (V.2 chemistry and 2 × 250 paired end sequencing), trimmed to 250 bp and quality filtered using default parameters in QIIME2 2019.10 [29]. Amplicon sequence variants (ASVs) were identified using Deblur within Qiime2, and taxonomy assigned by comparison to the Silva v123 database formatted for QIIME (see the electronic supplementary material for Data pipeline; NCBI SRA archive PRJNA387720; and Dryad Repository doi:10.5061/dryad.0zpc866vh [30] for resultant ASV table). Non-rarified data were used throughout and samples were compared using a multidimensional framework on ASV level taxonomic assignments. Nonmetric multidimensional scaling (nMDS) based on

Bray–Curtis similarity of log $(x + 1)$ transformed data was used to visualize differences among communities. Differences between sites were analysed using a PERMANOVA framework with sediment depths compared separately to maintain assumptions of sample independence. Statistical analyses were performed in PRIMER-e (ver.7; [31]).

Sequences identified to be within Euryarchaeota were isolated and compared with taxa from other seep sites globally and the ice-covered lakes of the McMurdo Dry Valleys. Representative sequences (operational taxonomic units based on 97% similarity identified through the same pipeline as in [27]) were compared to sequences from the literature from areas of known seepage; sequences were compiled from GenBank and VAMPS (SBJ_BME_Bv4v5 and SBJ_BME_Av4v5 from [32]). We used this approach to conservatively compare known seep sites to the CCS and allowed us to use established groupings rather than taxonomic assignments based on the databases alone. Aligned sequences (of 250 bp length) were constructed into trees using the BOOSTER (BOOTstrap Support by Transfer) platform within the GTR model of the phyML software (v. 3.0; [33,34]). The standard options within the BOOSTER platform were used, which included: 100 bootstrap replicates, a random starting tree with the SPR option, transfer bootstrap normalized supports in place of the classical Fleinstein approach, six substitution rate categories, and optimized topology, branch lengths, and rate parameters [34]. Constructed trees were re-rooted, and rotated when necessary in FigTree (v. 1.4.4). An analogous treatment of sulfate reducing bacteria was also performed following the same methods.

## 3. Results

### (a) Temporal observations

In 2012, the white filamentous microbial mat extended 70 m along the 10 m isobaths and consisted of discrete patches of white sediment, that microscopic inspection found to be composed of sheath forming bacteria analogous to the genus *Beggiatoa* common at methane seeps. The abundant seastar, *Odontaster validus*, was occasionally observed on the mat (figure 1). While additional cores were sieved on a 300 μm sieve, no macrofauna were observed within the sediment in 2012. In 2014, the site was visited and a single image identified the continued presence of the mat, however no samples were collected to characterize the microbial community. In 2016, the extensive mat retained its patchy white distribution but had been reduced in linear extent to 50 m. While cores were sieved for macrofauna again in 2016, only a single dorvilled polychaete was observed although adjacent to the mats, terebellid polychaetes were occasionally found during *in situ* sampling. During further exploration of the region, an additional linear area of seepage was observed at 7 m water depth that appeared to extend greater than 25 m in length. Owing to ice conditions, further exploration would have been unsafe so the full extent of this feature remains unknown. As we swam over this shallower site in 2012, and did not notice it, it is unlikely that the shallow (7 m water depth feature) was present at that time.

### (b) Sediment biogeochemistry

Methane in the 10 m water depth area (CCS) of seepage had an isotopic signature of $\delta^{13}C = -78 \pm 1‰$ and $\delta^2H = -361 \pm 16‰$ (mean ± standard error presented throughout manuscript unless otherwise indicated; n = 3). The 7 m water depth feature (the 'Shallow Site') had a divergent value

Proc. R. Soc. B 287: 20201134

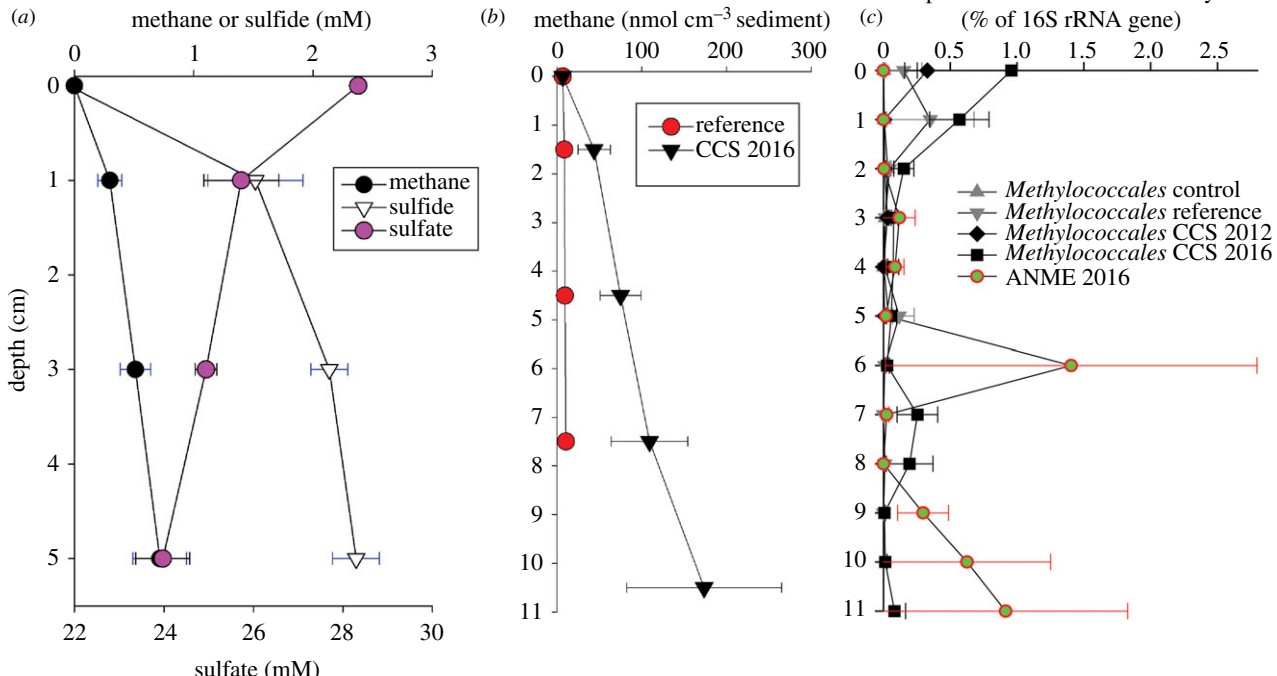

**Figure 2.** (*a*) The porewater at the Cinder Cones Seep showed non-complete exhaustion of sulfate with depth and no clear sulfate methane transition zone as measured from *in situ* porewater extraction in 2016 ($n = 3$). (*b*) In 2016, mean concentration of methane increased with depth within the Cinder Cones Seep and was below our detection limit in the reference site in 2016 ($n = 3$). Samples were from sediment plugs spanning 3 cm depth horizons ($n = 3$). (*c*) The vertical distribution of the relative per cent of rRNA genes belonging to taxa known to be methane oxidizers at the Cinder Cones methane seep. Cores were collected adjacent to porewater collected in (*a*), or from the same cores as methane concentrations in (*b*) and thus are a reduced dataset in contrast to figure 3. ANME, ANaerobic MEthane oxidizing archaea. No ANME were detected in 2012. All data show mean and standard error of three cores from each of the sites. (Online version in colour.)

$\delta^{13}C = -47 \pm 1\permil$ and a $\delta^2H$ of $-401\permil$. The concentration of methane within the CCS increased with increasing sediment depth up to $0.7 \pm 0.2$ mM methane in porewater at 5 cm sediment depth (figure 2*a*). When sediment plugs rather than *in situ* porewater extraction was used to quantify methane, the same pattern of increasing methane with depth was also observed; methane increased up to $0.4 \pm 0.1$ µmol $CH_4$ (sediment cm)$^{-3}$ at 12 cm (figure 2*b*). Overlying water methane concentrations were below our minimum reliable detection limit of 30 nM. The Shallow Site had methane present uniformly throughout the top 7 cm of the single core sampled with a concentration of $0.4$ µmol $CH_4$ (sediment cm)$^{-3}$. There was no clear sulfate-methane transition zone (SMTZ) present in any of the cores. The single non-quantitative, methane value measured from 2012 porewater was 32.6 mM $CH_4$, indicating that methane was present in 2012.

The ion and anion concentrations reflected shifting biogeochemistry with increasing depth that in many ways mirrored the pattern of methane (electronic supplementary material, table S1). At no point was sulfate fully depleted, ranging between 24 and 28 mM sulfate, with the lowest concentration measured at 5 cm depth (figure 2*b*). Sulfide was high, exceeding 2 mM at sediment depths from 1 cm to 5 cm and had an opposite distribution from sulfate and mirroring the depth pattern of methane. A sulfate–methane transition was not observed (figure 2*b*). A similar pattern was also present for nitrate, which decreased from the overlying water value of 13 µM to 1 µM $NO_3$ with depth. Ammonium increased with depth to 0.4 mM $NH_3$ at 5 cm depth within the sediment. Potassium, magnesium and calcium all varied by less than 0.5 mM with depth.

### (c) Methane flux

Along the 10 m depth feature, methane flux was measured as $3.1 \pm 0.9$ mmol methane m$^{-2}$ d$^{-1}$ ($n = 4$ chamber deployments) from the areas of seepage. The reference site had no measurable flux within our detection limit, meaning it never exceeded $0.02 \pm 0.02$ mmol $CH_4$ m$^{-2}$ d$^{-1}$ ($n = 2$ deployments; ± provides range).

### (d) Grain size

The sediment underlying the microbial mats consisted of larger grain sizes than adjacent reference sediment. The clearest shift was the sum of grains larger than 495 µm where $41 \pm 6\%$ of sediment were found in this fraction in the microbial mat compared to only $13 \pm 6\%$ within this size fraction at the adjacent control sites; there was a significant difference in the relative proportions of larger grains ($n = 3$ at each site, $t = 3.4$; $p = 0.027$; electronic supplementary material, figure S1). Microscopic examination revealed that basalt grains were interlaced with a crystalline matrix similar in appearance to calcite.

### (e) Microbial community

The relative abundance of ASVs identified as methane oxidizing taxa increased over the 5 years following the onset of seepage. The microbial community was characterized by greater than 1900 sequence of the V4 region of the 16S rRNA gene per sample post quality control (electronic supplementary material, Supplemental results). No ASVs of known ANME taxa were found in 2012 whereas in 2016, we recovered ASVs identified as ANME-1 in every core

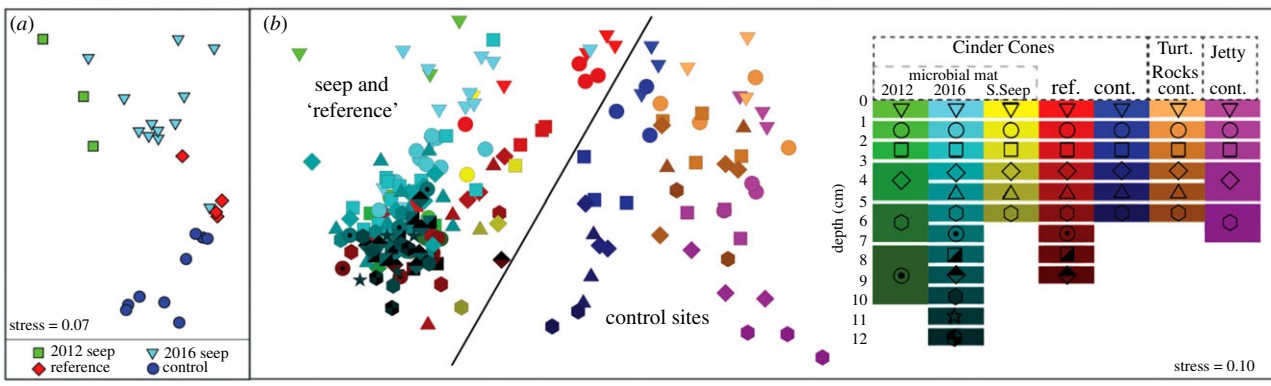

**Figure 3.** Non-metric multidimensional (nMDS) scaling to visualize microbial composition of the Cinder Cones methane seep (CCS) in comparison to reference (adjacent sediment) and control sites. (a) Surface sediment microbial composition demonstrating microbial shift over time at the CCS and in relation to control sites (all control sites are combined for this visualization). (b) nMDS showing all sites and depths sampled within the Ross Sea. Significant differences were present between control and both seep and reference sites. Statistical significance between sites are provided in the electronic supplementary material, table S2. S. Seep, Cinder Cones Shallow Seep; Turt. Rocks, Turtle Rocks; ref, reference site; cont, control site. (Online version in colour.)

taken from the microbial mat (figure 2c). This pattern was not driven by sequencing effort as sequencing depth was similar in both years, with greater than 9 k sequences post quality control for all vertical fractions within the CCS in both years. In 2016, these ANME-1 ASVs reached a maximum 4.1% of the microbial ASVs within the mat and were not found at the reference, Shallow Seep or control sites. ASVs identified as *Methylococcales*, an aerobic methane-oxidizing bacteria, increased in their proportional abundance from $0.33 \pm 0.17\%$ in 2012 to $0.83 \pm 0.53\%$ in 2016. *Methylococcales* ASVs were also present at the reference site, reaching a maximum of $0.35 \pm 0.37\%$ of the community at 1 cm sediment depth. No other known methanotrophs were identified at any of the sites (electronic supplementary material, Supplemental results and Discussion).

An analysis of the Euryarchaeota illuminated the diversity of ANME in the Cinder Cones habitat (electronic supplementary material, figure S2). Phylogenetic analysis identified four taxa from the 2016 CCS that clustered within branches that included other ANME-2 taxa from known seep and reducing habitats. The ANME-2 clustering taxa were rare, never making up more than 0.04% relative abundance of the microbial community. Three sequences fell within a branch that included ANME-3 taxa however that branch also included cultured *Methanococcoides* methanogens and we do not interpret those sequences as belonging to ANME-3. The three ANME-1 ASVs were more than 97% similar, and were included as a single operational taxonomic unit that fell within a relatively well supported branch of other deep-sea vent and seep ANME-1. The closest ANME-1 relatives were from the Pacific Ocean including Eel River and a more proximate seep in New Zealand.

When looking at the overall community composition, the microbial community within the CCS became increasingly dissimilar from the reference sediment community from 2012 to 2016. The surface sediment communities in both 2012 and 2016 were distinct from the nearby reference communities and each other (figure 3a, PERMANOVA one-way analysis, psuedo-$F_3 = 6.63$, $p = 0.001$; all pairwise comparisons were significant at $p < 0.05$). The overall trends were driven by an increase in the relative proportion of ASVs identified as sulfide-oxidizing bacteria (notably the non-mat forming *Sulfurovum* within the Campylobacterales) at the CCS compared to the reference and control sites and a decrease in

these same sulfide-oxidizing ASVs from 2012 to 2016 within the CCS (figure 4; electronic supplementary material, figure S3B). Below the surface sediment layer, the microbial community in 2012 was not different from the adjacent reference site (figure 3b, PERMANOVA results provided in electronic supplementary material, table S2). The control sites remained significantly different from all seep and reference samples at all sediment depths. ASVs of sulfate-reducing Deltaproteobacteria were the most common constituents across all sites and sediment depths. Included in this group were multiple clades known to form syntrophic associations with ANME (SEEP-1,-2, and -4 SRB; electronic supplementary material, figure S4). The Archaeal ASVs were dominated by the Woesearchaeota at all sites but also included members of the Asgard group at the CCS and reference sites, including Heimdallarchaeia, Lokiarchaeia and Odinarchaeia (figure 4). Methane cycling was also not limited to ANME, ASVs from four groups associated with methanogenesis were present, including Methanosarcinales, Methanofastidiosales, Methanomassiliicoccales, in addition to Bathyarchaeia.

## 4. Discussion

### (a) Biogeochemical underpinnings of the seep

This CCS provides a unique opportunity to understand the biogeochemistry of a methane seep that is actively developing, contrasting with our understanding of seep biogeochemistry based on developed seep sites. The geochemistry of this seep, while in many ways perplexing, is not surprising when we take into consideration that it is a 'new' seep. Methane porewater concentrations were lower than expected at less than 1 mM however flux was 21 g methane $m^{-2} d^{-1}$, which is similar to other seeps globally [35]. At established seeps, methane and sulfate are rarely found co-occurring as AOM exhausts the supply of sulfate and creates a sharp geochemical horizon within the sediment (known as the SMTZ). Changes in the depth of the SMTZ are driven by the rate of the AOM by ANME and can take years to centuries to stabilize following changes in methane seepage [36]. During multiple years of seepage in the CCS, ANME did not appear to be present and, thus, the driving force for the development of an SMTZ was absent. It is

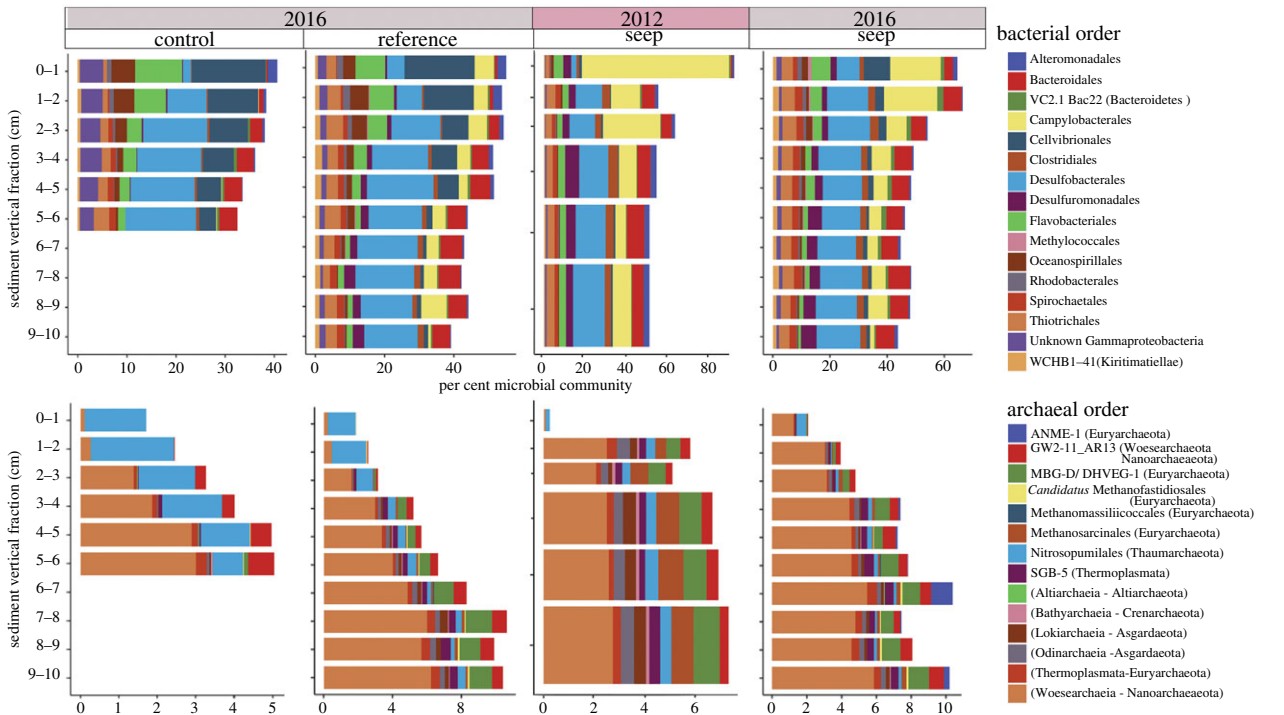

**Figure 4.** Per cent relative abundance of 16S rRNA genes identified as the indicated Bacteria (above) and Archaea (below) Orders in relation to vertical sediment depth across control ($n = 7$ cores at Turtle Rocks and Cinder Cones combined), the reference site (less than 2 m adjacent to seep; $n = 4$ cores), the Cinder Cones Seep itself in 2012 ($n = 3$ cores), and in 2016 ($n = 12$ cores). The 16 most dominant bacterial Orders, and 14 most dominant archaeal Orders are shown as they encompassed the dominant members of the community across all sites. (Online version in colour.)

only with ANME becoming established at the CCS that a clear SMTZ should be expected to develop, however that development could take multiple years to decades.

The similarity of the microbial community to the adjacent reference site also provides insight into the trajectory of microbial response to methane input. Following the onset of seepage certain taxa responded prior to sampling in 2012 (i.e. *Sulfurovum* at the sediment surface and Methanosarcinales below the surface; figure 4), however it took between 1 and 5 years for seepage to overcome the biological inertia of the microbial community and shift the community composition below the sediment surface quantitatively. This can be clearly seen by no significant difference in the microbial communities below the sediment surface between the reference and 2012 CCS samples, and a divergence between the microbial community at the reference and 2016 CCS microbial community between 0 and 5 cm sediment depth (figure 3b). It is also important to note that the relative proportion of ASVs identified as *Methylococcales* continued to increase at the sediment surface from 2012 to 2016, suggesting that their population continued to change as a result of methane input on a multiyear timescale. *Methylococcales* ASVs were also present at the reference site, although methane concentrations were below our detection limit (figure 2b). The high porosity of the sediment may allow periodic advective mixing of porewater from the CCS into the reference sites which could have also led to the observed similarity between the microbial communities. Regardless, it remains clear that the CCS microbial community took multiple years to adapt to seepage.

While we have been able to identify the source of methane as microbial and supporting evidence that the seep is associated with subsurface fluid flow, it is unclear

why the feature began seeping in 2011. Although the site itself occurs on the flank of an active volcano, stable isotopic analysis identified that the methane was produced by methanogenic archaea degrading an organic carbon source. When viewed in isolation, individual samples had $\delta^2$H and $\delta^{13}$C values that could indicate thermogenic methane (up to −320 and −48‰, respectively), however when both isotopes are viewed in concert all samples fell within the expected values for microbially produced methane (following [37]). Combining our porewater methane concentrations and methane flux, we calculate that fluid efflux from the sediment was $5.1\ \mathrm{l\ m^{-2}\ d^{-1}}$. This efflux suggests that the seep is fuelled by a significant subsurface fluid flow that advects in sulfide and methane to the microbial community. One hypothesis for this is that significant carbon burial led to fermentative methanogenesis in a quantity that was sufficient enough to, when paired with subsurface fluid flow, result in the formation of the seep structure. A logical source of this would be deposition of phytodetritus either from the annual phytoplankton bloom (*sensu* [38]) or significant input of ice algae and benthic diatom growth [39]; macroalgae is largely unknown from this region of McMurdo Sound and would be an unlikely cause.

We have no definitive geological explanation for the linear nature of both the 10 m and 7 m seep sites, yet cinder cone formation can often lead to diverse subsurface plumbing that could plausibly lead to the observed pattern of seepage. Cinder cones, normally formed as the result of explosive pyroclastic flows, can also form linear features of lava emission known as splatter ramparts or 'curtains of fire'. Following the active period of cinder cones, complex subsurface fissures can also form providing potential conduits for fluid flow that could manifest on the surface as

linear features [40]. The different grain size observed in the microbial mat compared to sediment directly adjacent is also congruent with the geological past leading to the surface manifestation of seepage. For now, the underlying cause for the conduit to become an area of active fluid flow in 2011 remains a mystery.

## (b) Microbial response to methane input: a model system

Antarctica is estimated to have a vast reservoir of methane trapped underneath ice sheets [5]. With increasing loss of ice sheets, such as the current retreat of the West Antarctic Ice Sheet [41], it is predicted that a significant volume of organic carbon could be released leading to both methanogenesis as well as the release of methane from subsurface reservoirs. This methane would be exposed to both benthic and pelagic communities that are largely naive to the input of methane. Upon the release of methane from under the melting ice sheets, a new microbial niche would open up and probably fuel methanotrophy in both aerobic and anaerobic forms. This model is what has been observed at the CCS, where the novel release of methane from a reservoir resulted in the input of a new electron donor to the sediment, leading to a shift in the microbial community. This discovery has allowed us to constrain, albeit on a coarse time scale, the rate in which methanotrophic communities can respond to methane release in the Antarctic.

We found that it took between 1 and 5 years for the microbial community to respond to the introduction of methane to the sediment. Estimated to consume between 70 and 90% of the methane that is released by benthic reservoirs, anaerobic methane-oxidizing archaea are among the most important microbes in the mitigation of the greenhouse gas on our planet [10]. These taxa, while slow growing, can be highly abundant within the sediment, for example making up (in addition to their sulfate reducing syntrophic partners) 30% of the microbial community and reaching population densities of $10^{10}$ microbes cm$^{-3}$ of the sediment [42]. In extreme cases, ANME have been shown to make up 50% of the total microbial community, although their abundance is often more in the range of 2–4% (of DNA) across a broad range of seep habitats [27]. In our samples, ANME ASVs were never more than 4.1% of the community, with few samples even reaching that proportion of the community and no ANME ASVs detected in the year 2012. With our sequencing depth (mean of 11 k sequence per depth horizon in 2012), if ANME were present in 2012 they made up less than 0.01% of the community. This result agrees with numerous other studies that have found it takes multiple years for microbial community to adapt, or begin to adapt, to the input of methane to the seafloor ecosystem [15,16].

## (c) Antarctic endemism, unknown taxa or early succession?

Surprisingly, the ANME lineages present were different than emerging biogeographic patterns would have suggested. The dominant ANME group within the CCS were ANME-1 with a small subset of the community being composed of ANME 2a-b lineages (max = 0.04%). ANME-1 ASVs were present in every sediment core from the CCS in 2016. However, these taxa are thought to be less adapted to cold temperatures than ANME-2 [43] and are rarely the dominant ANME group [10]. Furthermore, one of the global dominant taxa ANME-2c (found at 83% of seeps; [10]), that we would have expected at the observed methane flux rates [44], was exceedingly rare (a total of four sequences out of 1.4 M). In addition, ANME-3, the dominant taxa at high latitude (the dormant Larson B seep; [21] and the HMMV; [22]) and cold-temperature seeps [45] were absent. This raises two potential possibilities for our observations: (i) there are taxa consuming methane that do not fall within the known groups of ANME as speculated by Saxton et al. [32]; or (ii) we are at an early successional stage of the microbial community.

The Antarctic is typified by a high proportion of endemic fauna, however, the impact of this on microbial biogeography is not completely known. The Drake Passage opened 30+ million years ago resulting in a strong oceanographic barrier to dispersal. This, combined with the subsequent cooling of the continent, has resulted in a high proportion of fauna found in the Antarctic being endemic at the genus level [46,47]. This same selection may have led to unique microbial groups, including methane oxidizing taxa that are currently not known from elsewhere. Within the McMurdo Dry Valleys, Saxton et al. [32] found the biogeochemical fingerprint of anaerobic methane oxidizing taxa but no known ANME lineages. In comparing Euryarchaeota between the CCS and the Dry Valley site, we found similar taxa including branches that appeared to fall within the Thermoplasmata and Methanomicrobiales. While different primers were used, and Archaea and rarer members of the community are especially sensitive to primer choice [48], the potential for endemic high latitude, Antarctic methanotrophs is provocative. Continued investigations at this site may help disentangle the role of novel methanotrophs and potential endemism versus ecological succession on the microbial community structure observed.

Microbial communities that take multiple years to adapt to methane release from Antarctica and the Southern Ocean may impact the role of Southern Ocean methane in global atmospheric forcing. A guiding principle in microbial ecosystems, known as the Baas-Becking hypothesis, is that 'everything is everywhere and the environment selects', however, we rarely consider the time scale that this environmental selection takes, even though this time scale could be on the order of years or more. When we traditionally model the impact of methane on the atmosphere we assume that the microbial oxidation of methane is complete and thus methane release will just increase the amount of $CO_2$ into the atmosphere (sensu [5]). We show here that the response of the microbial ecosystem is not rapid enough to validate this assumption. We found that it took years for the beginnings of a methanotrophic community to develop in response to novel methane release.

The assumption that methanotrophs will respond rapidly to novel methane release is not supported across a variety of other ocean ecosystems including other sites in the Southern Ocean. Methane release from South Georgia island was not oxidized completely in the sediment [19]. Further, the water column dynamics of the Bransfield Straight lead to this region being a net source of methane into the atmosphere [49]. If we use shallow water habitats from the Arctic as a model [50], significant methane flux from the sediment into

the atmosphere is possible. Here, we found that methane release on the order of $3 \, \mathrm{mmol \, m^{-2} \, d^{-1}}$ continued after a minimum of 5 years after methane seepage began and highlights the importance of microbial succession in determining the magnitude of methane release in future climate scenarios. Including this variable in future modelling studies, such as the landmark work of Wadham *et al.* [5], would probably allow for better prediction of the role of the Southern Ocean in the global methane cycle and future global change.

## (d) Summary

Here, we describe the formation and development of a novel methane seep in the High Antarctic and quantify the evolution of the microbial community over a 5 year time period. We found that it took up to 5 years for microorganisms capable of forming a methane 'sediment filter' to develop. In this time period we observed a sequential shift in the microbial community to a group of taxa that were unexpected based on the temperature, biogeochemical environment, and location. We also observed a continued release of methane out of the sediment surface after this time period. While the ultimate source of this methane remains unknown, the ability of the CCS to inform our understanding of microbial succession and to predict the magnitude of methane release from our oceans in response to warming and ice shelf retreat are significant. Although we focus on one particular area, the Ross Sea is an exciting area of methane research with observed bottom simulating reflectors indicative of methane hydrates present just north of our study site [51]. Our results suggest that the accuracy of future global climate models may be improved by considering the time it will take for microbial communities to respond to novel methane input.

Data accessibility. DNA Sequence: GenBank SRA Archives PRJNA387720. Final Amplicon Sequence Variant Table: Dryad Digital Repository: https://doi.org/10.5061/dryad.0zpc866vh [30]. Porewater Solute Composition: BCO-DMO doi:10.1575/1912/bco-dmo.770638.1.

Authors' contributions. A.R.T., S.S. and R.M.W. collected samples, processed those samples and wrote the manuscript. A.R.T. and S.S. prepared figures and did all data analysis. Funding for this project was obtained by A.R.T.

Competing interests. We declare we have no competing interests.

Funding. This work was supported by the National Science Foundation Office of Polar Program grants 1642570 and 1103428.

Acknowledgements. This work was made possible owing to information provided by Dr Stacy Kim who found the site. We thank Rob Robbins and Steve Rupp for their support of this research as well as the community of McMurdo station. We are indebted to R. Vega Thurber, A. Pasulka and V. Orphan for sample analysis and guidance on this work.

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
