## [Reviewer comments · Proceedings of the Royal Society B: Biological Sciences]

Review History

RSPB-2019-2007.R0 (Original submission)

Review form: Reviewer 1

Recommendation

Major revision is needed (please make suggestions in comments)

Scientific importance: Is the manuscript an original and important contribution to its field?

Excellent

General interest: Is the paper of sufficient general interest?

Good

Quality of the paper: Is the overall quality of the paper suitable?

Acceptable

Is the length of the paper justified?

Yes

Should the paper be seen by a specialist statistical reviewer?

No

Do you have any concerns about statistical analyses in this paper? If so, please specify them explicitly in your report.

No

It is a condition of publication that authors make their supporting data, code and materials available - either as supplementary material or hosted in an external repository. Please rate, if applicable, the supporting data on the following criteria.

Is it accessible?

Yes

Is it clear?

Yes

Is it adequate?

Yes

Do you have any ethical concerns with this paper?

No

Comments to the Author

The manuscript by Thurber et al. on the shallow water methane seepage in the Ross Sea is well written and is of global interest in the context of climate change and biogeochemical cycling. The discoveries of methane seepage in Southern Ocean continental shelf habitat is very recent and this study provides the first measurements of methane fluxes next to the description of a newly started methane seepage system in shallow waters.

This manuscript will be useful for experts and modellers in climate change and biogeochemical cycling as well as for specialists in microbiology, biodiversity, ecology and geology/hydrofluid circulation in the Southern Ocean.

The authors might not be aware of the published cruise report of R/V METEOR Cruise Report M134, Emissions of Free Gas from Cross-Shelf Troughs of South Georgia: Distribution, Quantification, and Sources for Methane Ebullition Sites in Sub-Antarctic Waters, Port Stanley (Falkland Islands) - Punta Arenas (Chile), 16 January - 18 February 2017 by Bohrmann et al. 2017 which is available online <https://elib.suub.uni-bremen.de/edocs/00106081-1.pdf>. Further recent discoveries of either methane flares (Spain et al. 2019) or high methane concentrations in marine sediments (de Valle et al. 2017) were made on the Kerguelen Plateau and eastern Antarctic Peninsula.

I have only a few points and suggestions for the revision which are listed below.

Introduction:

Line 72: Southern Ocean should be in capital.

Line 74: Please edit to "East Scotia Ridge hydrothermal vents fields" as the two vent fields discovered by Rogers et al. 2012 are on the segments E2 and E9 of the East Scotia Ridge and about 440km apart from each other.

Line 75: Instead of "South Georgia Island habitat" I recommend to write "South Georgia continental shelf habitat"

Line 76ff: Please rephrase the sentences as the seep did not find clams. While Niemann et al. 2009 cited the large vesicomid clams as belonging to the genus Calyptogena, this genus has undergone significant revision and the erection of new genera and it might be appropriate in this manuscript to indicate this. The large vesicomid clam species, formerly often placed into the genus Calyptogena, are all members of the subfamily Pliocardiinae.

L 77: Genus names like *Calyptogena* should be in italics.

Material and Methods:

Please give the versions and references for the software used like for Primer-e vers.7 , phyML, R or FigTree.

Add a section on the grain size analysis method and give reference for the phi size scale used.

Line 164, 235: What taxon do you mean with Euryarchaea? You refer to Euryarchaeota in lines 165 or in Figure legend 3.

Results:

In the results section you refer to "Temporal Observations" and show in-situ images in Figure one. While the figure legend refers to bacterial mats and white mats are seen on the images, the result text does not mention these. Please add a sentence on the white mats to the results, describing the "feature" (line 174) in more detail. These images also show vagile Antarctic benthic fauna, especially a red starfish resembling the genus *Odontaster*. While this manuscript is not describing the mega-, macro- or meiofauna associated with the methane seepage, as at least one species is seen on and near the white mats, this should be mentioned in the temporal observations of the habitat.

Sediment biogeochemistry:

Refer to figure 2 A& B as well as Supplement Table 1 in the text.

In the paragraph please be clear if your 10 m and 7m depth refer to water depth or sediment depth.

Line 180: Delete "both" in the sentence. I suggest splitting the sentence and then in the first part of the paragraph report on the biogenic origin of the methane, defined by its isotopic signature. The second part of the paragraph then reports on the concentration increase with vertical depth. Line 184 ff: In the section on the methane increase with increasing depth please indicate if a sulfate-methane transition zone was detected or not.

Grain size:

Mention how many cores have been analysed for grain size and what type of grains were found next to the basalt grains and crystalline structures. Could these crystalline structures be either calcite or ikaite or glendonite (See Peckmann 2017 *Geology* 45 (6):575-576). Are the crystalline structures one shape and size?

Microbial Community:

Mention how many cores have been analysed for the microbial analysis.

Figure 2D is reported on before 2C, I suggest change the order of the figures.

Please state clearly to which figure (main or supplement) you refer when describing the microbial community in the text and use the same taxonomic names in the text and figure so a reader can refer to them. For example the text mentions *Methylococcales* but these are not mentioned in any figure. Supplement figure 2 on the Archaea is not reported on at all.

Line 219: Genus *Sulforovum* in italics

Phylogeny:

Please reword this section and remove phrases like "most surprising" and "additional surprise". Be clearer in describing the phylogenetic relatedness. For example describe your ANME-1 precisely in the text; is it the sequence marked as Cinder Cones Seep - 2016 - ANME-1? Are the groups ANME-1s , 2s and 3 defined by you or established groupings?

Nothing is said about the node support of the tree.

Line 235: Do you mean Euryarchara or Euryarchaeota?

Line 240: Genus *Methanolobus* in italics

Discussion:

Microbial Response to Methane Input: A Model System:

This study's results on the slow establishment of anaerobic methane-oxidizing microbes in Southern Ocean marine sediments after the start of methane seepage are important in the context of current climate change. The current climate change might have the potential to cause methane release in the Southern Ocean and around Antarctic caused by changes in ice covered and sea temperatures.

Antarctic Endemism, Unknown Taxa, or early Succession?

Like in the results section of the Phylogeny, it is not clear in this section, which ANME groups you refer to your own sequenced ones or established groupings. Please reword this section and name your sequenced groups as they appear in Figure 3.

Line 358: Please name the emerging paradigms that you mention.

Figure 1:

Can you please add a scale bar to the four images. The starfish give a rough reference of scale between the images.

Figure 2:

I suggest swapping C and D to follow their appearance in the results.

Figure 3:

Can you please add information on node support to the tree.

Supplement Table 1

No comments

Supplement Figure 1

Please explain in the legend text that the grain size is shown in phi scale. In the legend in the figure it would be advisable to show $\phi > 0$, $\phi = 0-0.5$, $\phi = 0.5-1$, $\phi = 1-3$, $\phi = 3-4$. Initially I thought the grain size was given in mm as the " ϕ - size" is placed well above the legend.

Supplement Figure 2

I suggest to have the taxon legends for Bacteria and Archaea to the right of the plots, not within any plot.

The legend for the Archaea does not show orders, it includes different taxonomic levels of Archaea, spanning different phyla, classes and orders. Please indicate at least the phylum the taxonomic unit belongs to. This enables readers that are less familiar with the Archaea to assess the presence and relative abundance of Euryarchaeota.

Review form: Reviewer 2

Recommendation

Major revision is needed (please make suggestions in comments)

Scientific importance: Is the manuscript an original and important contribution to its field?

Excellent

General interest: Is the paper of sufficient general interest?

Good

Quality of the paper: Is the overall quality of the paper suitable?

Marginal

Is the length of the paper justified?

Yes

Should the paper be seen by a specialist statistical reviewer?

No

Do you have any concerns about statistical analyses in this paper? If so, please specify them explicitly in your report.

No

It is a condition of publication that authors make their supporting data, code and materials available - either as supplementary material or hosted in an external repository. Please rate, if applicable, the supporting data on the following criteria.

Is it accessible?

Yes

Is it clear?

Yes

Is it adequate?

Yes

Do you have any ethical concerns with this paper?

No

Comments to the Author

See attachment

Decision letter (RSPB-2019-2007.R0)

28-Oct-2019

Dear Dr Thurber:

I am writing to inform you that your manuscript RSPB-2019-2007 entitled "Riddles in the Cold: Antarctic Endemism and Microbial Succession impact methane cycling in the Southern Ocean" has, in its current form, been rejected for publication in Proceedings B.

This action has been taken on the advice of referees, who have recommended that substantial revisions are necessary. With this in mind we would be happy to consider a resubmission, provided the comments of the referees are fully addressed. However please note that this is not a provisional acceptance.

Sincerely,

Dr Daniel Costa
 mailto: proceedingsb@royalsociety.org

Associate Editor
 Board Member: 1
 Comments to Author:

In two very detailed reviews, both reviewers raise multiple points and recommend major revision. I agree with the reviewer's comments, which should be absorbed carefully into the revised version. Both reviewers also comment on the quality of the writing at present. The revision needs to incorporate the vast majority of the reviewer suggestions and be written far more carefully and tightly.

Reviewer(s)' Comments to Author:

Referee: 1

Comments to the Author(s)

The manuscript by Thurber et al. on the shallow water methane seepage in the Ross Sea is well written and is of global interest in the context of climate change and biogeochemical cycling. The discoveries of methane seepage in Southern Ocean continental shelf habitat is very recent and this study provides the first measurements of methane fluxes next to the description of a newly started methane seepage system in shallow waters.

This manuscript will be useful for experts and modellers in climate change and biogeochemical cycling as well as for specialists in microbiology, biodiversity, ecology and geology/hydrofluid circulation in the Southern Ocean.

The authors might not be aware of the published cruise report of R/V METEOR Cruise Report M134, Emissions of Free Gas from Cross-Shelf Troughs of South Georgia: Distribution, Quantification, and Sources for Methane Ebullition Sites in Sub-Antarctic Waters, Port Stanley (Falkland Islands) - Punta Arenas (Chile), 16 January - 18 February 2017 by Bohrmann et al. 2017 which is available online <https://elib.suub.uni-bremen.de/edocs/00106081-1.pdf>. Further recent discoveries of either methane flares (Spain et al. 2019) or high methane concentrations in marine sediments (de Valle et al. 2017) were made on the Kerguelen Plateau and eastern Antarctic Peninsula.

I have only a few points and suggestions for the revision which are listed below.

Introduction:

Line 72: Southern Ocean should be in capital.

Line 74: Please edit to "East Scotia Ridge hydrothermal vents fields" as the two vent fields discovered by Rogers et al. 2012 are on the segments E2 and E9 of the East Scotia Ridge and about 440km apart from each other.

Line 75: Instead of "South Georgia Island habitat" I recommend to write "South Georgia continental shelf habitat"

Line 76ff: Please rephrase the sentences as the seep did not find clams. While Niemann et al. 2009 cited the large vesicomid clams as belonging to the genus *Calyptogena*, this genus has undergone significant revision and the erection of new genera and it might be appropriate in this manuscript to indicate this. The large vesicomid clam species, formerly often placed into the genus *Calyptogena*, are all members of the subfamily *Pliocardiinae*.

L 77: Genus names like *Calyptogena* should be in italics.

Material and Methods:

Please give the versions and references for the software used like for Primer-e vers.7 , phyML, R or FigTree.

Add a section on the grain size analysis method and give reference for the phi size scale used.

Line 164, 235: What taxon do you mean with Euryarchaea? You refer to Euryarchaeota in lines 165 or in Figure legend 3.

Results:

In the results section you refer to "Temporal Observations" and show in-situ images in Figure one. While the figure legend refers to bacterial mats and white mats are seen on the images, the result text does not mention these. Please add a sentence on the white mats to the results, describing the "feature" (line 174) in more detail. These images also show vagile Antarctic benthic fauna, especially a red starfish resembling the genus *Odontaster*. While this manuscript is not describing the mega-, macro- or meiofauna associated with the methane seepage, as at least one species is seen on and near the white mats, this should be mentioned in the temporal observations of the habitat.

Sediment biogeochemistry:

Refer to figure 2 A& B as well as Supplement Table 1 in the text.

In the paragraph please be clear if your 10 m and 7m depth refer to water depth or sediment depth.

Line 180: Delete "both" in the sentence. I suggest splitting the sentence and then in the first part of the paragraph report on the biogenic origin of the methane, defined by its isotopic signature. The second part of the paragraph then reports on the concentration increase with vertical depth.

Line 184 ff: In the section on the methane increase with increasing depth please indicate if a sulfate-methane transition zone was detected or not.

Grain size:

Mention how many cores have been analysed for grain size and what type of grains were found next to the basalt grains and crystalline structures. Could these crystalline structures be either calcite or ikaite or glendonite (See Peckmann 2017 *Geology* 45 (6):575-576). Are the crystalline structures one shape and size?

Microbial Community:

Mention how many cores have been analysed for the microbial analysis.

Figure 2D is reported on before 2C, I suggest change the order of the figures.

Please state clearly to which figure (main or supplement) you refer when describing the microbial community in the text and use the same taxonomic names in the text and figure so a reader can refer to them. For example the text mentions *Methylococcales* but these are not mentioned in any figure. Supplement figure 2 on the Archaea is not reported on at all.

Line 219: Genus *Sulforovum* in italics

Phylogeny:

Please reword this section and remove phrases like “most surprising” and “additional surprise”. Be clearer in describing the phylogenetic relatedness. For example describe your ANME-1 precisely in the text; is it the sequence marked as Cinder Cones Seep – 2016 – ANME-1? Are the groups ANME-1s, 2s and 3 defined by you or established groupings?

Nothing is said about the node support of the tree.

Line 235: Do you mean Euryarchara or Euryarchaeota?

Line 240: Genus *Methanolobus* in italics

Discussion:

Microbial Response to Methane Input: A Model System:

This study's results on the slow establishment of anaerobic methane-oxidizing microbes in Southern Ocean marine sediments after the start of methane seepage are important in the context of current climate change. The current climate change might have the potential to cause methane release in the Southern Ocean and around Antarctic caused by changes in ice covered and sea temperatures.

Antarctic Endemism, Unknown Taxa, or early Succession?

Like in the results section of the Phylogeny, it is not clear in this section, which ANME groups you refer to your own sequenced ones or established groupings. Please reword this section and name your sequenced groups as they appear in Figure 3.

Line 358: Please name the emerging paradigms that you mention.

Figure 1:

Can you please add a scale bar to the four images. The starfish give a rough reference of scale between the images.

Figure 2:

I suggest swapping C and D to follow their appearance in the results.

Figure 3:

Can you please add information on node support to the tree.

Supplement Table 1

No comments

Supplement Figure 1

Please explain in the legend text that the grain size is shown in phi scale. In the legend in the figure it would be advisable to show $\phi > 0$, $\phi = 0-0.5$, $\phi = 0.5-1$, $\phi = 1-3$, $\phi = 3-4$. Initially I thought the grain size was given in mm as the “ ϕ - size” is placed well above the legend.

Supplement Figure 2

I suggest to have the taxon legends for Bacteria and Archaea to the right of the plots, not within any plot.

The legend for the Archaea does not show orders, it includes different taxonomic levels of Archaea, spanning different phyla, classes and orders. Please indicate at least the phylum the taxonomic unit belongs to. This enables readers that are less familiar with the Archaea to assess the presence and relative abundance of Euryarchaeota.

Referee: 2

Comments to the Author(s)

See attachment

Author's Response to Decision Letter for (RSPB-2019-2007.R0)

See Appendix A.

RSPB-2020-1134.R0

Review form: Reviewer 1

Recommendation

Accept as is

Scientific importance: Is the manuscript an original and important contribution to its field?

Excellent

General interest: Is the paper of sufficient general interest?

Excellent

Quality of the paper: Is the overall quality of the paper suitable?

Good

Is the length of the paper justified?

Yes

Should the paper be seen by a specialist statistical reviewer?

No

Do you have any concerns about statistical analyses in this paper? If so, please specify them explicitly in your report.

No

It is a condition of publication that authors make their supporting data, code and materials available - either as supplementary material or hosted in an external repository. Please rate, if applicable, the supporting data on the following criteria.

Is it accessible?

Yes

Is it clear?

Yes

Is it adequate?

Yes

Do you have any ethical concerns with this paper?

No

Comments to the Author

I thanks the authors for the thorough revision and resubmission of their manuscript and have no further comments other than I am looking forward to seeing it published.

Decision letter (RSPB-2020-1134.R0)

11-Jun-2020

Dear Dr Thurber

I am pleased to inform you that your manuscript RSPB-2020-1134 entitled "Riddles in the Cold: Antarctic Endemism and Microbial Succession impact methane cycling in the Southern Ocean" has been accepted for publication in Proceedings B.

The referee(s) have recommended publication, but also suggest some minor revisions to your manuscript. Therefore, I invite you to respond to the referee(s)' comments and revise your manuscript. Because the schedule for publication is very tight, it is a condition of publication that you submit the revised version of your manuscript within 7 days. If you do not think you will be able to meet this date please let us know.

[http://datadryad.org/submit?journalID=RSPB&manu=\(Document not available\)](http://datadryad.org/submit?journalID=RSPB&manu=(Document%20not%20available)) which will take you to your unique entry in the Dryad repository. If you have already submitted your data to dryad you can make any necessary revisions to your dataset by following the above link. Please see <https://royalsociety.org/journals/ethics-policies/data-sharing-mining/> for more details.

Sincerely,

Dr Daniel Costa

Associate Editor

Comments to Author:

Thank you for addressing the previous reviewing comments, and the considerable effort which has gone into revision, which are now satisfactorily resolved.

Reviewer(s)' Comments to Author:

Referee: 1

Comments to the Author(s).

I thank the authors for the thorough revision and resubmission of their manuscript and have no further comments other than I am looking forward to seeing it published.

Author's Response to Decision Letter for (RSPB-2020-1134.R0)

See Appendix B.

Decision letter (RSPB-2020-1134.R1)

26-Jun-2020

Dear Dr Thurber

I am pleased to inform you that your manuscript entitled "Riddles in the Cold: Antarctic Endemism and Microbial Succession impact methane cycling in the Southern Ocean" has been accepted for publication in Proceedings B.

Open Access

You are invited to opt for Open Access, making your freely available to all as soon as it is ready for publication under a CC BY licence. Our article processing charge for Open Access is £1700. Corresponding authors from member institutions (<http://royalsocietypublishing.org/site/librarians/allmembers.xhtml>) receive a 25% discount to these charges. For more information please visit <http://royalsocietypublishing.org/open-access>.

Paper charges

Sincerely,
Editor, Proceedings B
<mailto:proceedingsb@royalsociety.org>

Appendix A

'response to referees'

Associate Editor

Board Member: 1

Comments to Author:

In two very detailed reviews, both reviewers raise multiple points and recommend major revision. I agree with the reviewer's comments, which should be absorbed carefully into the revised version. Both reviewers also comment on the quality of the writing at present. The revision needs to incorporate the vast majority of the reviewer suggestions and be written far more carefully and tightly.

Thank you for the input and we have gone through and addressed and in almost every case followed the reviewers suggested edits. Both reviewers provided comprehensive comments and corrections that we feel have further improved the manuscript. We have also carefully evaluated the writing to clean and tighten up the flow and information. This has been changed throughout and is clear from the "track changes" version of the manuscript. Following is a line by line identification of how the manuscript was changed following reviewer comments. Notably, as a result of reviewer comments we have remade nearly all figures, redone all statistical tests, expanded the supplemental material greatly, added additional datasets and data analysis pipelines for transparency.

We agree with reviewer comments concerning what may appear as a cursory treatment of our dataset. We did this purposefully to tell one specific story focusing on a particular group that is globally important. Even with this streamlined version of our story, we are still at the word and page limit for Proceedings of the Royal Society - Biology. To address these comments, we have added a supplemental results and discussion section, as well as a further three figures (two of which are supplemental) in the aim to directly address the concerns that our treatment of our dataset was not in depth.

Following a reviewer comment we have also now shifted from an Operational Taxonomic Unit (OTU) approach following the standard 97% similarity to identify a taxa to a more up to date method (Deblur) to generate Amplicon Sequence Variants (ASVs). This has not impacted the results but has resulted in our remaking all of our figures. In addition, we have re-done our phylogenetic trees using recently released pipelines that allow more robust estimation of tree branch support based on short read sequences. Shifting to ASVs rather than OSUs did not impact the results significantly, however the new tree approach did change the structure of our phylogenetic trees and we have modified the results and discussion correspondingly.

A final point that we would like to address globally is a lack of making our data available - we had made our data available in raw form through the NCBI Archive and NSF repositories and are providing a full ASV table for all samples as supplemental so people have the greatest access to our data possible. For review we are including this in supplemental data but our aim for the full submission is to include this in the Dryad Repository for which ProcB has partnered for this purpose. We agree with the reviewers that data transparency is critical and we hope that these steps are acceptable to the journal. We also now provide, in supplemental, a full data analysis pipeline so anyone can recreate our analyses exactly.

Reviewer(s)' Comments to Author:

Referee: 1

Comments to the Author(s)

The manuscript by Thurber et al. on the shallow water methane seepage in the Ross Sea is well written and is of global interest in the context of climate change and biogeochemical cycling. The discoveries of methane seepage in Southern Ocean continental shelf habitat is very recent and this study provides the first measurements of methane fluxes next to the description of a newly started methane seepage system in shallow waters. This manuscript will be useful for experts and modellers in climate change and biogeochemical cycling as well as for specialists in microbiology, biodiversity, ecology and geology/hydrofluid circulation in the Southern Ocean.

The authors might not be aware of the published cruise report of R/V METEOR Cruise Report M134, Emissions of Free Gas from Cross-Shelf Troughs of South Georgia: Distribution, Quantification, and Sources for Methane Ebullition Sites in Sub-Antarctic Waters, Port Stanley (Falkland Islands) - Punta Arenas (Chile), 16 January - 18 February 2017 by Bohrmann et al. 2017 which is available online <https://elib.suub.uni-bremen.de/edocs/00106081-1.pdf>. Further recent discoveries of either methane flares (Spain et al. 2019) or high methane concentrations in marine sediments (de Valle et al. 2017) were made on the Kerguelen Plateau and eastern Antarctic Peninsula.

I was unable to find the methane flare paper by Spain et al. 2019, we have now included de Valle et al. 2017 in the reference list in addition to updating the references with a few exciting papers that have recently come out concerning the Antarctic methane cycle.

We have now also added reference to the Meteor Cruise Report. Thank you for providing that information.

I have only a few points and suggestions for the revision which are listed below.

Introduction:

Line 72: Southern Ocean should be in capital.

Fixed throughout^[at1]

Line 74: Please edit to "East Scotia Ridge hydrothermal vents fields" as the two vent fields discovered by Rogers et al. 2012 are on the segments E2 and E9 of the East Scotia Ridge and about 440km apart from each other.

We have removed this sentence while streamlining the introduction.

Line 75: Instead of "South Georgia Island habitat" I recommend to write "South Georgia continental shelf habitat"

We have removed this sentence while streamlining the introduction.

Line 76ff: Please rephrase the sentences as the seep did not find clams. While Niemann et al. 2009 cited the large vesicomyid clams as belonging to the genus *Calyptogena*, this genus has undergone significant revision and the erection of new genera and it might be appropriate in this manuscript to indicate this. The large vesicomyid clam species, formerly often placed into the genus *Calyptogena*, are all members of the subfamily Pliocardiinae.

We have removed this sentence while streamlining the introduction.

L 77: Genus names like *Calyptogena* should be in italics.

We have removed this the reference to *Calyptogena* while streamlining the introduction, however have italicized all Genera referenced in the MS now.

Material and Methods:

Please give the versions and references for the software used like for Primer-e vers.7 , phyML, R or FigTree.

We have done this throughout. This is most noticeable at L132-144.

Add a section on the grain size analysis method and give reference for the phi size scale used.

We have put in a brief description of methods and modified the figure to show size range rather than Phi scale. Please see L119 and Supplemental Figure 1.

Line 164, 235: What taxon do you mean with Euryarchaea? You refer to Euryarchaeota in lines 165 or in Figure legend 3.

> We have swapped all euryarchaea to euryarchaeota following this comment (in both text and figure legends) throughout.

Results:

In the results section you refer to "Temporal Observations" and show in-situ images in Figure one. While the figure legend refers to bacterial mats and white mats are seen on the images, the result text does not mention these. Please add a sentence on the white mats to the results, describing the "feature" (line 174) in more detail. These images also show vagile Antarctic benthic fauna, especially a red starfish resembling the genus *Odontaster*. While this manuscript is not describing the mega-, macro- or meiofauna associated with the methane seepage, as at least one species is seen on and near the white mats, this should be mentioned in the temporal observations of the habitat.

> This has been added - in addition, we had analyzed the feature for macrofauna and there essentially wasn't any. I have made a note of this in the paper now and reference that *Odontaster validus* was present on it. Please see a completely rewritten section L165-178

Sediment biogeochemistry:

Refer to figure 2 A& B as well as Supplement Table 1 in the text.

In the paragraph please be clear if your 10 m and 7m depth refer to water depth or sediment depth.

This has been fixed through out and we found this a confusing point and have since shifted to referring to the 10m Seep as the Cinder Cones Seep and the 7m seep the Shallow Site. We now specify this on lines 181 to 184.

Line 180: Delete "both" in the sentence. I suggest splitting the sentence and then in the first part of the paragraph report on the biogenic origin of the methane, defined by its isotopic signature. The second part of the paragraph then reports on the concentration increase with vertical depth.

We have followed this good suggestion, however the next reviewer felt it was inappropriate to discuss that it was biogenic based on its signature within the results so we have moved that discussion out of the results.

Line 184 ff: In the section on the methane increase with increasing depth please indicate if a sulfate-methane transition zone was detected or not.

We have added a line to the methane section identifying a lack of SMTZ. L199

Grain size:

Mention how many cores have been analysed for grain size and what type of grains were found next to the basalt grains and crystalline structures. Could these crystalline structures be either calcite or ikaite or glendonite (See Peckmann 2017 *Geology* 45 (6):575-576). Are the crystalline structures one shape and size?

We have done XRD as well as microscopy of the grains and they appeared to be basaltic grains with what appeared to be magnesium calcite. We felt that this aspect of the seep could be dealt with more fully in a separate manuscript with directed approach (and expertise beyond the authors). We had analyzed three cores from each of the regions for grain size. We have put in what information we feel confident about without more specific analysis (L214). In addition, we clarified that we have three replicates at both Reference and Seep sites. L214

Microbial Community:

Mention how many cores have been analysed for the microbial analysis.

This has now been added into the manuscript in the methods, as well as discussed in detail in the supplemental methods and we now provide a table (Supplemental Table 3) that further clarifies this point. Methods clarification can be found at L98 and 102. Within Supplemental discussion is focused in L8-15 but largely found in Supplemental Table 3.

Figure 2D is reported on before 2C, I suggest to change the order of the figures.

Following this point, we have restructured the results to discuss 2C before 2D and in addition moved 2D to a new figure (Figure 3) following reviewer suggestions to show data beyond the surface sediment. This also allowed us to match up the y-axis of chemical data and microbial data.

Please state clearly to which figure (main or supplement) you refer when describing the microbial community in the text and use the same taxonomic names in the text and figure so a reader can refer to them. For example the text mentions Methylococcales but these are not mentioned in any figure. Supplement figure 2 on the Archaea is not reported on at all.

We have done this throughout the manuscript and in the figure legends. When discussing certain taxa, we also identify what taxonomic grouping it fits into in the figure (for example *Sulforovum* is not shown in figure 4, so we identify that it is in the Campylobacterales so it is clear how that is represented in the figures).

Line 219: Genus *Sulforovum* in italics

Done.

Phylogeny:

Please reword this section and remove phrases like “most surprising” and “additional surprise”. Be clearer in describing the phylogenetic relatedness. For example describe your ANME-1 precisely in the text; is it the sequence marked as Cinder Cones Seep – 2016 – ANME-1? Are the groups ANME-1s , 2s and 3 defined by you or established groupings?

We now clarify that we used established taxa based on the literature for the tree construction (L150-154). We have also reworded the section entirely.

Nothing is said about the node support of the tree.

Based on this comment we remade the trees using the Booster platform to provide node support and also potentially improve the robustness of the tree. Please note that this has changed some of the topology of the tree as it required a complete re-analysis. The text has been modified to reflect this change and the resultant tree is less clear, so we have moved it to supplemental material.

Line 235: Do you mean Euryarchara or Euryarchaeota?

This has been fixed.

Line 240: Genus Methanolobus in italics

We have removed this sentence.

Discussion:

Microbial Response to Methane Input: A Model System:

This study’s results on the slow establishment of anaerobic methane-oxidizing microbes in Southern Ocean marine sediments after the start of methane seepage are important in the context of current climate change. The current climate change might have the potential to cause methane release in the Southern Ocean and around Antarctic caused by changes in ice covered and sea temperatures.

Antarctic Endemism, Unknown Taxa, or early Succession?

Like in the results section of the Phylogeny, it is not clear in this section, which ANME groups you refer to your own sequenced ones or established groupings. Please reword this section and name your sequenced groups as they appear in Figure 3.

We have rewritten this section to clarify this point, and also reduced it based on the new tree that resulted from the above comment (See lines 350).

Line 358: Please name the emerging paradigms that you mention.

I believe we may have overstated that it was a paradigm, but in particular we were stating that the taxa present were not expected in the cold and high sulfate environment that we sampled. We have rephrased paradigm to “emerging biogeographic patterns”

Figure 1:

Can you please add a scale bar to the four images. The starfish give a rough reference of scale between the images.

We do not have a way to provide an accurate scale bar for these images. We could provide the mean size of sea stars at the site in 2016 however we could only provide that for 2016 and any population size shifts across the years could lead to potential confusion.

Figure 2:

I suggest swapping C and D to follow their appearance in the results.

We have moved D to Figure 3.

Figure 3:

Can you please add information on node support to the tree.

We have now added node support information in the tree, as well as the SRB tree in the supplemental methods. Supplemental Figure 2 and 4

Supplement Table 1

No comments

Supplement Figure 1

Please explain in the legend text that the grain size is shown in phi scale. In the legend in the figure it would be advisable to show $\phi > 0$, $\phi = 0-0.5$, $\phi = 0.5-1$, $\phi = 1-3$, $\phi = 3-4$. Initially I thought the grain size was given in mm as the “ ϕ - size” is placed well above the legend.

We have changed the scale to be in μm to remove any ambiguity. This figure is still Supplemental Figure 1.

Supplement Figure 2

I suggest to have the taxon legends for Bacteria and Archaea to the right of the plots, not within any plot.

We have followed this suggestion. This is now figure 4 in the main text.

The legend for the Archaea does not show orders, it includes different taxonomic levels of Archaea, spanning different phyla, classes and orders. Please indicate at least the phylum the taxonomic unit belongs to. This enables readers that are less familiar with the Archaea to assess the presence and relative abundance of Euryarchaeota.

We have redone this figure and now provide both Order and Phyla for all the Archaea that were abundant. Please see figure 4.

Reviewer 2

This manuscript reports on a very interesting discovery of a new methane-sulfide seep reporting on the biogeochemistry and sediment-associated microbial community at the seep and in nearby control locations. The initial findings presented here are an important finding for high latitude, particularly Antarctic, ecosystems. I suspect that the article will be of sufficiently broad interest to the Proceedings of the Royal Society B's readership. The discussion is the strongest part of the manuscript, it presents interesting interpretations and description of the the ecosystem as the authors have interpreted it.

We thank the reviewer for their praise and in particular the discussion. We also thank this (and the previous reviewer) for such comprehensive comments and guidance.

Despite the importance of this work, the presentation of the results and writing of the manuscript warrant further work to best present the significance of this and

communicate the message to the reader. Emphasizing the importance of this work at high latitudes is needed, and formulating it in the perspective for how the present work advances understanding of the methane cycle first in the Antarctic, then examining the relevance to global processes will improve the story presented. For example Lines 85-88: “To better understand the future role of methane in the coupled ocean-earth system, two fundamental questions remain: (1) What is the rate at which microbial communities can respond to changes in the methane cycle; and (2) Do Antarctic methane seep communities function similarly to those elsewhere in the globe” doesn’t quite capture the importance of this work – and is easy to argue with as far as other potential fundamental questions.

We have rephrased and re-written the introduction to better argue the importance of this research and also tried to better phrase the work considering this. While we felt those two points in the referenced sentence did highlight the importance we also see how they could lack specificity and have modified them accordingly (Please see L75-77)

Environmental descriptive information was sparsely reported though this is critical for understanding the system and how this is different from others. Specific points are detailed below. In addition to the point raised above, the low temperature nature and importance of this site at ~ -2C compared to deep sea sites at 4C is important to the response of these very slow growing methane producing and oxidizing processes – although this point was not raised in the manuscript. This is an issue I believe pertaining to the point raised above and differentrates at which communities respond.

We have now added specific reference the temperature of the habitat L87. We had not emphasized this as the temperature is not that unique when compared to the Håkon Mosby Mud Volcano and Chilean seeps (the later are at 1.8 C).

The presentation of the molecular work is at a high level, this limits the impact of the results presented. The presentation should meet the standard in the field and clearly justify the results that are presented, which are a fraction of the data collected.

We had chosen to focus on a specific aspect of the data set to tell a focused story in a relatively short format journal. In response to this comment we have now greatly expanded the discussion of taxa, generated a series of new (including multipanel figures), and expanded our statistical analysis. As part of this we have written a supplemental discussion and results to better discuss the datasets. In

addition, we have put in a much expanded discussion of the results in the manuscript proper and include two figures, one previously from the supplemental (remade) and a new one that allows a greater view of the depth. In generating these figures, we also increased our resolution of separating the sites and now define a Reference site, that was previously considered part of the control.

To fit this in the manuscript we have greatly streamlined the introduction and large portions of the methods.

(i) There was no mention in the manuscript of potential sulfate reducing bacterial partners of the anaerobic methane oxidizers, despite the large data set of bacterial diversity data collected.

This is a significant issue with the manuscript as it stands.

In response to this comment we have now included brief mention of the SRB taxa that we found, however in the supplemental we present a tree to show the diversity of taxa likely responsible for sulfate reduction including those that fall within SRB SEEP clades (based on literature sequence). As with many aspects of large data sets such as those generated, it is likely that a comprehensive discussion of the SRB taxa could form a stand alone paper.

(ii) Were oxygen analyses conducted with the porewater samples? The presence of both sulfur and methane oxidizing bacteria at depth in the sediments should be discussed.

Oxygen analysis was plagued by the high porosity of the sediment leading to oxygenation of the water prior to analysis. We were not able to do in situ microprofiling work and our external microprofiling led to erroneous and irreproducible oxygen profiles. We now identify this at L112.

(iii) Methodological limitations for the molecular detection work were not mentioned at all, but we now know that especially for low biomass representatives of the microbial community, as is often the case with archaea, that the results are sensitive to primer sets being used (e.g. Fisher et al. 2016 *Frontiers in Microbiology*, 7:1297. doi: 10.3389/fmicb.2016.01297). It's possible that relative abundances of iTag sequences could be different with different primers (Saxton et al. 2016 and others use a variety of primer sets to reach the depth of Archaea).

We have now raised this concern in the manuscript (L 372) and have downplayed our treatment of the comparisons, including moving the phylogenetic tree to supplemental.

(iv) The authors may want to consider analysis of the data so that amplicon sequence variants (ASVs which are high quality sequences) are reported rather than OTUs which are sequence clusters. Going forward this will allow for other investigators to compare their sequences directly with ASVs reported here. This has become the standard by many in the field.

We have followed this reviewer's suggestion, redone all analyses using Deblur, and shifted our discussion to ASVs instead of OTUs. We have redone our community analysis on unclustered (i.e. raw ASVs). We had already made our data available through the SRA archive, which will make our data as useful as possible for future comparison.

(v) I would argue for inclusion of Supplemental Fig. 2 in the manuscript (with a more informative legend either way that describes the values that are actually represented, and with a color palette that can be visually interpreted). It would seem that given the data transparency policy of this journal, it might be useful to provide the full OTU or ASV tables as supplemental information, or at least on BCO-DMO.

> We have followed this advice and moved the previous supplemental figure 2 from supplemental figure 2 to Figure 4 in the manuscript. We also now provide a full ASV table as supplemental documentation associated with this submission including sample site information. For review, we include this as supplemental but if this manuscript is accepted for publication we aim to submit these data to the Dryad repository, which has partnered for ProcB for this purpose.

(vi) The title has “microbial succession” in it, which to me suggests that there would be a consideration of the community as a whole; perhaps the title should be modified to succession of anaerobic methane oxidizers?

> We appreciate this comment and made a significant effort to streamline the message considering the data sets that are incorporated in this manuscript. In response to this

we have added a more comprehensive community discussion in addition to comparison across sediment depths and now include figure 3B which shows shift (or in some cases lack of shift) in the microbial community. We discuss community shifts throughout, including sulfide oxidizers and similarity across the years (for example the paragraph that starts at L279). Further we also discuss this in the supplemental results and discussion and notably in Supplemental figure 3. When working at the ASV level, it also makes approaches such as SIMPER less informative, or at least more overwhelming, than when working on higher taxonomic levels, however we did use those approaches and used them in an exploratory approach to guide our discussion rather than state the results explicitly.

(vii) Was the white putative sulfur oxidizing mat actually sampled for DNA sequencing? Not clear; the text mentions (lines 222-224) mat-forming taxa were present, but at low levels 2.7-3.6%; and increased, while the mat density decreased.

Attempts to sequence the mat in particular were unsuccessful in identifying a bacterial taxa known to be mat forming. Beggiatoa was present but in very low abundance and not uniformly present in samples with abundant mat present. We decided to not specifically discuss this as it would be largely conjecture based on those data we generated.

(viii) What is the evidence for endemism? The “most abundant ANME group” was ANME-1s appears to have identical sequences to others from North Pacific and New Zealand (note that data supporting that claim are in the results; Line 360-361).

We have greatly downplayed this however still feel it has a place in the discussion to guide future work. Please see lines 363 - 375

Detailed suggestions for improving the manuscript follow:

1. The challenges of responding to such an event could be better communicated, as this impacts the science conducted.

> We entirely agree however have not changed the manuscript to respond to this comment. It is entirely pertinent to the outcome but we struggle to incorporate a

response that does not come across as complaining about rejected proposals or the reduced funding received as this was considered a high risk project.

2. The abstract writing should be tighter, and better reflect the findings in a more quantitative manner. It creates more questions than it answers.

- Suggest replacing the words “in which” in place of where in the first sentence

We replaced the first sentence while tightening up the manuscript (L19).

- Qualify which marine habitats you are referring to in the second sentence, line 15

Done. L17

- The sentences starting line 17 really should be specific to what this study addresses; e.g. This study reports on a new H₂S and CH₄ seep at high latitude where the findings contribute to better understanding of ...

Modified to follow this suggestion. L20

- Line 19-20: Suggest adding some more quantitative information such as mat size, seafloor depth, methane concentrations etc., to the sentence

We have done this.

- Line 19 – 20: revise the point that the mat was discovered in 2012, but formed in 2011...

Done. L19

seems that it was discovered in 2011 and sampled in 2012. Are there pictures from 2010, or 2011?

> These images do not exist but we felt that as written in the original manuscript this was misleading and have rewritten the abstract to remove this lack of clarity. L11

- Line 23: suggest adding “at high latitude” at the end of the sentence; this is the very unique aspect of this work.

Done. L20 (although we added it in the middle of the sentence.)

- Line 26: wording “ANME had become present, however in low numbers and an unexpected lineage for the physical and biogeochemical... “ is awkward. They were

detected. Mention the unexpected lineage... not sure we are so good at microbial ecology to expect lineages.

We have identified who was found and why it was unexpected.

- The last sentence begs the question of what evidence is there that was not seeping methane in this system previously? The manuscript states work has been done in the region (presumably there are some phototransects of the same area – pre-seep?), but the evidence is not shown in the manuscript.

We hope that the last sentence of the abstract will entice the reader to read further, as this point - which is very important - is beyond what we could fit within the character limit of the abstract.

3. Introduction:

- Line 34: suggest adding the following after the words yet methane, “produced biogenically and thermogenically,” is an integral part...

> We have not followed this suggestion as we feel that, while an important point, is too specific for an opening sentence in the manuscript.

- Line 37: suggest adding the concentrations that the methane has increased to since 1750.

>Done Please see L 38

- Line 41: suggest adding the words “and Southern Ocean” after Antarctica – the Southern Ocean is not the same as Antarctica.

> Done carefully throughout. Work at South Georgia and other places have advanced our knowledge of the Southern Ocean methane cycle, and the real unknown is the Antarctic Continent itself (or adjacent islands that are heavily ice influenced, such as Ross Island). The methane volumes are for the continent not the Southern Ocean. As a result, we have evaluated each use of Antarctica vs Southern Ocean throughout the manuscript and adjusted as appropriate.

-Line 43 the word “or” should be changed to “and”

> Sentence was entirely rephrased.

- Lines 51-58: in general what proportion of methane in the ocean is thought to be oxidized aerobically? This would be interesting to add to the background information provided.

> Added in “the majority of methane not oxidized by ANME” L 62 as estimates are almost everything released in deep water, most in shallow but not everything but likely most. We do not feel comfortable based on what data are available to provide a universal estimate for this.

- Line 57: amend the information presented such that Betaproteobacteria as well as Verrucomicrobia are also aerobic methane oxidizers.

> We have added a qualifying statement in the referenced sentence as “The majority of methane not oxidized by ANME aggregates is aerobically oxidized by a diversity bacteria including Methylococcaceae (γ -Proteobacteria) in marine systems.” L61 We have also added to the supplemental discussion that neither of the alphaproteobacteria taxa known to oxidize methane were present (at the family level) in the samples. The Verrucomicrobia methanotrophs are thought to be acidophilic and we found no pattern across our sites suggesting they were involved in methane cycle. In addition other taxa, including the Betaproteobacteria and NC10 phyla (the later nitrifying) were both absent. This is also included in supplemental discussion and we now present a plot of the Verrucomicrobia within the system in supplemental results and discussion to show that their distribution was unlikely tied to methane seepage. We have left the singular reference in to the Methyococcales as it appears (at least currently) to be the most universal marine methanotroph and specifically introduces what we found in the system.

- Line 60: clarify who is “they” that are being referred to.

> Sentence was largely changed making this request mute, but we went through to make sure there were no similar mistakes in the manuscript.

-Line 64: would be interesting to add the location/site of this study to the sentence for comparison to the present study.

> I greatly cut down on the information in this sentence due to word limitations and to tighten up writing. We have not provided a map as the samples were along the length of the transect inclusive of both ends and the middle. Please see Lines 98-100 which reads “Sampling points were randomly distributed within patches of white, putative sulfur-oxidizing filamentous bacteria in 2012 (n=6) and 2016 (n=12) along the feature and purposefully including both ends of the feature to capture along feature variance. “

- Line 67: Suggest that methane content could be better phrased as volume of the methane reservoir.

> Paragraph heavily modified. Percent of reservoir now given in abstract and specified in first sentence.

-Line 68: Microorganisms are referred to sometimes (especially in host-associated systems as flora); but not as fauna. In this sentence and elsewhere (line 92) just calling them out as microbiota, or microorganisms makes sense.

> Done throughout.

-Line 71: wording “emission a” should be emissions and

> Sentence deleted

- Lines 79-83: the wording here is awkwardly stated.

> Sentence deleted

- Line 89: wording “on the Antarctic Continent” could be enhanced, to include on the flanks of volcanic Ross Island, the depth, and latitude longitude better than 78 degrees S would help the reader.

> We have modified the sentence to say flanks of volcanic Ross Island, removed the Lat, however since we provide lat long and depth in the following “Site Description” we have not expanded further. Please see :78 “seep at a site known as Cinder Cones in McMurdo Sound within the Ross Sea” and L83 “The Cinder Cones Seep (CCS) is on the flanks of the volcanic Ross Island (77° 47.998' S 166° 40.241' E). “

- Line 90: might as well say what the area has been studied for as I was wondering if it was the outfall site.

> Different sites. The Outfall is around past Winters Quarters Bay and directly in front of the station. This is on the north side of the Hut Point peninsula, closer to Castle Rock. We added that the ongoing research was due to an Ice burg scour being present. Honestly, we are not sure why it was chosen for research in the 60s other than there is a persistent crack there and it was likely as far east as one can easily go without crossing a significant pressure ridge. We have added that the recent studies there were driven by an ice burg scour study (L 89-90)

- Line 91: remove the word this before 2011.

Sentence Reworded.

4. Methods:

-Lines 102-103: clarify whether any photographs have been taken of the site prior to 2012 (as suggested in Fig. 1).

> Sadly no. This site is often visited but no one had noticed this feature - which would be almost impossible to miss. In addition, the person who had most recently been diving there frequently (Dr. Stacy Kim) has worked extensively at hydrothermal vent systems and so would have noticed the microbial mat. We have reworded this section to try and make this clear. L88 The mat was not seen in 2010 despite being a prominent feature when it appeared in 2011 and occurring at a site studied since the mid-1960s for its ecology, including as a site of an ice burg scour at deeper depths [25].

-Line 110: suggest changing haphazardly to random; and white microbial mats could be better described as white, putative sulfur oxidizing filamentous microbial mats.

>Thank you for the suggestion - this has been done.

-Lines 113-116: I found the description confusing; 3 cm sub cores taken initially prior to 1 cm subsectioning?

We modified the sentence to make it clearer, the cores had a subcore taken while vertically slicing.

-Line 124: insert a comma after “that”

This section was highly reduced in length which also deleted this sentence.

-Line 128: clarify how many cores were collected for microbial characterization in 2012 and in 2016. Were the actual locations of the samples recorded – not clear. *It would be helpful to have a supplemental map/figure with the coring locations including reference sites for 2012 and 2016. This could be quite helpful to sort out the differences and cores for biogeochemistry vs. microbial characterization, vs. flux measurement sites.

The cores used for figure 2 were from the two ends and the center of the site and correspond to the biogeochemical data (figure 2). We now make this clear in the figure legend. Number of cores in the different sampling years is now clearly specified in the methods section as this was not clear and we provide increased specificity in Supplemental Table 3.

-Line 141: what kind of benthic chambers (model, citation, square area?)

These were custom designed and built by us. We have included the square area sampled.

-Line 146: how was all the sediment within the chamber removed? To what depth?

This has now been included. These are done by diver so we dug underneath the side and capped the bottom and pulled them out of the mud. L129 “Upon recovery, the entire chamber was removed, including the sediment within it, by capping the bottom of the chamber and extracting it from the sediment including approximately the top 10cm of sediment.”

-Line 151: not clear what depths the samples for microbial characterization were sampled at? (could clarify that too in the preceding paragraph).

We now have this in in the sediment collection area by referring to Figure 4, which shows it clearly. There were different sampling depths at different years, and for statistical comparison (Supplement Table 2) we binned samples to allow cross comparison and we provide even more clarity on the replication in Supplemental table 3.

-Line 158: clarify this sentence, which is not currently a sentence.

Sentence deleted.

-Line 153: remove “of”

Sentence reworked.

-Lines 151-158: Add a few more details such as how many tags per sample were sequenced (depth); and were these data normalized between samples? What was the method used?

Sequencing depth is now specified (>1900) per sample, and much more information provided in supplemental (especially Supplemental Table 3). We did not normalize between samples as we ran all of our analyses binned at various levels and it made no significant difference to the results.

-Lines 163, or thereabouts: I would recommend making a statement that justifies the selection of taxonomic orders used in the bar charts (Fig. 2d and Supp. Fig. 2); these appear to be cherry picked – but that selection should be justified here.

For Fig 2d (now 2C) we include all taxa with a clear affinity to methane oxidation. We now include a statement in the supplemental results about the other taxa that could have been included but were not found (alpha-and beta-proteobacteria) and a plot demonstrating that if the distribution of the Verrucomicrobiales is not likely supportive of methane oxidation. For Suppl Fig 2, we chose the top taxa that were abundant in either the Seep or the control cores, and still provided useful

information. We used an arbitrary cut off for the number of taxa to include based on iterative exploration of the data on what showed patterns in both seep and non-seep sites. We now explain this in the figure legend. This was unclear, and we thank the reviewer for pointing this out. We have specified in (2C) figure legend that this was all fauna that are putative methanotrophs.

-Line 164: Suggest here clarifying the method used for finding neighboring sequences (identified by Blast or assigned by Bassien classifier etc.).

We carefully combed the literature for ANME and Seep SRB signature as we were concerned that blasting would lead to environmental data that may or may not actually be ANME or Seep SRB. We then used BOOSTER to create the trees and provides the bootstrap values.

-Line 168: might add that all sequences were trimmed to 250 base pairs.

Added. Please see line 155.

Results:

- Line 174: qualify the “feature” as white filamentous microbial mat...

Done. L166

- Lines 183-184; interpretations of methane source should be in the discussion, not results section.

We have modified this sentence to only include results. This has now been moved to the discussion.

- Supp. Fig. 2: are averages of iTag relative occurrences shown for all cores taken in 2016 for all the selected orders? Explain this further in the text, and legend. (this was referred to above).

We now add in the figure legend that it was the average of the cores collected. This is now in Figure 4 and we specify the sample site and number of cores analyzed, in certain cases samples did not pass QC in sufficient quantity to be included and we also provide depth specific replication values in Supplemental Table 3.

Interpret this figure – the dominant Order found is Wosearchaeota – however this or was not even mentioned.

We have more fully interpreted this figure both within the results as well as the supplemental discussion now.

- Fig 2a and 2b were never cited; the legend doesn't state whether the data represents 2012 or 2016 cores.

They were from 2016. We now site them and also have the years indicated in the legend.

- Lines 200-202 the nitrate values in the table are in mmol should that be mM? While the text says that the values decrease from 13-1 micromolar; this is not apparent in the table – values are 0.14 to 0.13 mmol.

The table has been fixed to show mM. See Supplemental Table 1

- Lines 209-213 It's not quite clear that the interpretation of Supplemental Fig. 1 is accurate; the numbers referred to do not jive with what is shown in the figure. In fact, the grain size in the reference sample looks to be quite a bit larger than the microbial mat sample.

This was a bit confusing due to using the Phi scale for the figure. We have just changed the figure legend to using the size of grains in um to make it clear that the microbial mat samples did have a greater proportion of their grains in larger size classes.

- Line 217 – Change to Fig. 2d. I suggest that the plots for all depths, or plots for different depths are made available as supplemental material; that the 0-1 depth is shown is perhaps not quite as interesting (should be more or less oxygenated) as deeper in the sediments. Though all the references are shown in green in the figure, were all from 2016? If not – maybe distinguish those between 2012 and 2016. I would suggest reporting Bray-Curtis similarities when comparing the reference to 2012 to 2016 community structures.

We have followed this advice and broken figure 2d into figure 3, including both the surface sediment (A) which shows a really clear pattern, and B which shows all depths and the clear separation of the Seep, Reference, and Sites. We also include a more comprehensive statistical analyses of these samples included as a supplemental table. We do not report Brey-Curtis Similarity as we feel this new treatment is easier to visualize the difference among the different samples.

- Line 219: seems like it would be useful to actually show the Sulfurovum data, and how that varies in proximity to the seep.

We now include a vertical plot of this taxa (as well as other key taxa) as Supplemental Figure 3.

-Line 222: Camplobacterales is an order; though the text says Camplybacteria. Stick with the use of Order.

Thank you. This has been fixed.

-Line 222-224: it's possible that there are other mat forming bacteria were in the mix; I suggest re-evaluating the data set for filamentous taxa.

It is absolutely possible that there are mat forming taxa that were not identified as such based on our analysis. We fully expected to find a significant proportion of the community as Thiotrichales. Beggiatoa were amazingly sparse in the data set. We omitted a further discussion in the manuscript as it rapidly led to conjecture. Even focused sequencing on what was mostly mat resulted in a diverse microbial community that led to no clearer answers.

-Line-224: were the gamma proteobacteria Methylococcales the only aerobic methane oxidizers identified out of the community?

They were the most unequivocal methane oxidizers present. We now more fully discuss this in the supplemental methods. None of the Beta or Alpha proteobacteria that are likely candidates were present. I fully expect that there were more present, however none were apparent.

- Fig 2c: there are no error bars in this plot – how were the data calculated, and what is being shown? The x-axis title should not be % of Microbes... it's relative percent of rRNA gene occurrence?

We have modified this figure legend to reflect that it is the relative percent of rRNA gene occurrence and included error bars.

-Lines 228-233 and in the Phylogeny paragraph: suggest proper word usage “No ANME were present in 2012”... be specific about what is being detected – that these are representative sequences that are, or were not detected. What is being detected are sequences, not organisms per se. This may seem minor, but it's an important distinction.

We have fixed this here and throughout. We now discuss the distribution of ASVs rather than taxa following this recommendation.

-Line 240: change to blast “against” what database at GenBank

We have removed this section.

-Fig. 3: The figure is not readable as it is; and is in poor resolution even if zoomed in. In the legend describe the use of shading and text colors. In addition placeholder “known” archaea should be added to the parts of the tree that are sparsely populated.

We have entirely redone this tree and also moved it to supplemental. We believe resolution was lost as it was turned in to a PDF by the journal as the resolution of the uploaded figure was high quality.

-Lines 244-246: Describe this result further; what defines a “clade” and how closely related in reality were the Lake Fryxell sequences to the CCS Methanomicrobiales sequences that are putative anaerobic methane oxidizers?

This paragraph has been heavily edited as through providing more robust bootstrap support the shape of the tree changed dramatically.

Discussion:

Some of the above comments will influence the discussion, and interpretation – so those will need to be followed through.

-Line 335: this would be a nice place to report the Bray-Curtis similarity value.

Due to the various depths and sequences, we have decided not to report this metric but rely on the expanded supplemental results and discussion to more fully address this.

-Line 337: reword “at least four years” to between 1-5 years, to make that accurate with the study.

Done L 133

-Lines 378-381: The distance between the Lake Fryxell and CCS sequences for the Fryxell groups is ~ 8-10%, which is pretty high; were these the nearest neighboring sequences when compared to Silva database; look at this more thoroughly and discuss.

We have omitted this section.

-Lines 389-396: clarify differences in marine vs. terrestrial processes; it’s clear that methane is not totally consumed in the terrestrial arctic. Methane is also produced in the

surface ocean by non-archaeal methanogenic pathways that can provide methane to the atmosphere (Repeta et al. 2016, Nat. Geosci. 9:884-887).

We have modified this sentence but not delved completely into the methane cycle.

-Line 414: change cause to source.

Done.

In summary, we again would like to thank the reviewers for the significant time they invested to improve the manuscript.

Appendix B

Response to Reviews:

Author Response: The Associate Editor and Referee accepted the manuscript (Comments pasted below). I have gone through and added the Media Summary, updated the data repository to include files now stored within Dryad, and added the required Data accessibility section. The Supplemental has been combined into a single document and updated to reflect the required modifications (Journal Name, doi, etc.). I have also added a few potential web images/ cover images and added them to the submission. I would like to thank the Associate Editor and the Referee for their time and insight. Changes have been tracked in the document below.

“Associate Editor

Comments to Author:

Thank you for addressing the previous reviewing comments, and the considerable effort which has gone into revision, which are now satisfactorily resolved.

Reviewer(s)' Comments to Author:

Referee: 1

Comments to the Author(s).

I thank the authors for the thorough revision and resubmission of their manuscript and have no further comments other than I am looking forward to seeing it published.”